# Early Life Factors Influencing Children Gut Microbiota at 3.5 Years from Two French Birth Cohorts

**DOI:** 10.3390/microorganisms11061390

**Published:** 2023-05-25

**Authors:** Gaël Toubon, Marie-José Butel, Jean-Christophe Rozé, Ioannis Nicolis, Johanne Delannoy, Cécile Zaros, Pierre-Yves Ancel, Julio Aires, Marie-Aline Charles

**Affiliations:** 1Centre de Recherche en Épidémiologie et StatistiqueS (CRESS), Inserm, INRAE, Université Paris Cité et Université Sorbonne Paris Nord, 75004 Paris, France; 2Physiopathologie et Pharmacotoxicologie Placentaire Humaine Microbiote Pré & Postnatal (3PHM), Inserm, UMR-S 1139, Université Paris Cité, 75006 Paris, France; 3FHU PREMA, Fighting Prematurity, 75014 Paris, France; 4Physiologie des Adaptations Nutritionnelles (PhAN), INRAE, UMR 1280, Université Hospitalière de Nantes, 44093 Nantes, France; 5EA7537 Biostatistique, Modélisation et Traitement des Données Biologiques (BioSTM), Université Paris Cité, 75006 Paris, France; 6Ined, Inserm, EFS Joint Unit Elfe, 93322 Aubervilliers, France

**Keywords:** gut microbiota, preschool children, early life factors, DOHaD, enterotypes

## Abstract

Early life gut microbiota-influencing factors may play an important role in programming individuals long-term health and substantial efforts have been devoted into studying the development of the gut microbiota in relation to early life events. This study aimed to examine in a single study, the persistence of associations between 20 factors occurring in the early life and the gut microbiota at 3.5 years of 798 children from two French nationwide birth cohorts, EPIPAGE 2 (very preterm children) and ELFE (late preterm and full-term children). Gut microbiota profiling was assessed using 16S rRNA gene sequencing-based method. Upon thorough adjustment of confounding factors, we demonstrated that gestational age was one of the factors most associated with gut microbiota differences with a noticeable imprint of prematurity at 3.5 years of age. Children born by cesarean section harbored lower richness and diversity and a different overall gut microbiota composition independently of preterm status. Children who had ever received human milk were associated with a *Prevotella*-driven enterotype (P_type) compared to those who had never received human milk. Living with a sibling was associated with higher diversity. Children with siblings and those attending daycare centers were associated with a P_type enterotype. Maternal factors including the country of birth and preconception maternal body mass index were associated with some microbiota characteristics: children born to overweight or obese mothers showed increased gut microbiota richness. This study reveals that multiple exposures operating from early life imprint the gut microbiota at 3.5 years that is a pivotal age when the gut microbiota acquires many of its adult characteristics.

## 1. Introduction

Due to its possible effects on health and diseases, the study of the human microbiota has become a research area of utmost interest in the past years [1]. The gut microbiota plays an important role in different aspects of physiological human traits including regulation of the intestine homeostasis, role in digestion and metabolism processes, immune system stimulation, brain–gut communication, and brain development [2]. Given these numerous physiological functions, gut dysbiosis has been studied in terms of association with a variety of non-communicable diseases, such as obesity, type 2 diabetes, inflammatory bowel disease, allergies, autism, and some types of cancer [2]. The developmental origins of health and disease (DOHaD) theory states that exposures during the highly plastic period of early life, the so called “*first thousand days period*”, can program later life disease risk [3]. Within the DOHaD paradigm, the gut microbiota has emerged as a potential critical early-life factor able to program long-term health [4]. Accordingly, alterations of the early gut microbiota, and therefore perturbations of the crosstalk between the gut microbiota and its functions set the stage for potential lifelong adverse outcomes [5]. From birth onwards, the development of the gut microbiota is regulated by a complex interplay between multiple exposures including maternal factors, diet, and environment [6]. Extensive data exist in the literature regarding the gut microbiota development for infants (less than 3 years); however, the latter has been overlooked in preschool (3–6 years) children [7] and particularly in the preterm population. The gut microbiota is purportedly shifting toward an adult-like microbiota around the age of 3 years [8] but that age has been revised upward recently (approximately 6 years) with the matter still being under debate [9,10,11]. It is crucial to fill the gap of knowledge during this important phase of life when the foundation of the relatively stable and resilient adult type gut microbiota is setting [7]. In this study, we leveraged the EPIPAGE 2 and ELFE French nationwide birth cohorts. We aimed to investigate in one single study including a large pediatric population of children born preterm or full-term, whether a variety of exposures operating from early in life (gestation to 1 year of life), still influenced the gut microbiota of children at 3.5 years of age. We reported the data obtained taking into account selected factors known to influence the gut microbiota that were categorized in different domains including maternal health, perinatal factors, diet, and environmental factors. Our findings may help to evaluate which early life factors might represent potential levers to modulate the children gut microbiota and propose opportunities for microbiota-targeted health-promoting strategies early in life.

## 2. Materials and Methods

### 2.1. EPIPAGE 2 and ELFE Cohorts

Children included in this study are part of two French national birth cohorts launched in 2011; EPIPAGE 2 (Etude Epidémiologique sur les Petits Ages Gestationnels-2) [12], and ELFE (Etude Longitudinale Française depuis l’Enfance) [13]. Briefly, EPIPAGE 2 is a cohort of exclusively preterm infant born before 34 weeks of gestational age (GA) recruited in 68 neonatal intensive care units (NICUs) in France. For a subset of EPIPAGE 2 preterm infants born before 32 weeks of gestational age recruited in 24 voluntary NICUs (EPIFLORE ancillary study of EPIPAGE 2), a collection of stool samples was carried out. ELFE is a nationwide birth cohort, which included newborns born after 33 weeks of gestational age in 349 randomly selected maternity units in mainland France. For a subset of these children, stool samples were collected at 3.5 years of age.

### 2.2. Ethics

The cohorts and the collection of stools were approved by the Committee for the Protection of People Participating in Biomedical Research (Comité de Protection des Personnes [CPP]), the national advisory committee on information processing in health research (Comité Consultatif sur le Traitement de l’Information en matière de Recherche dans le domaine de la Santé [CCTIRS]) and the French National Data Protection Authority (Commission Nationale de l’Informatique et des Libertes [CNIL]). Recruitment and data collection occurred only after the families had received information and agreed to participate in these cohorts with informed consent.

### 2.3. Participants

Our ancillary study included 208 and 630 children from EPIPAGE 2 and ELFE, respectively.

As twin children shared a more similar gut microbiota (see pairwise beta diversity distances between related and unrelated children and genus composition among related children in Appendix A), we randomly excluded one of the twins from a related pair or two children from triplets to assess as much variability as possible in the gut microbiota of children at 3.5 years of age, leading to the non-inclusion of 40 children (EPIPAGE 2, *n* = 36; ELFE, *n* = 4). The final dataset on which the analyses were performed included a total of 798 children (EPIPAGE 2, *n* = 172; ELFE, *n* = 626).

### 2.4. Maternal and Child Data Collection

For both cohorts, information regarding pregnancy, newborn and maternal characteristics were collected using parental questionnaires at birth, during follow-up, or extracted from the obstetrical file (both cohorts), and medical files during the perinatal period (EPIPAGE 2).

### 2.5. Exposures of Interest

For the current study, we defined the early exposure window as the first year of life of the child. For both cohorts, we selected factors occurring during this exposure timeframe that have a potential influence on the child gut microbiota. These factors included gestational age, delivery mode, exposure to antibiotics within the first 3 days of life (referred to as early neonatal antibiotherapy), human milk consumption, skin-to-skin practice, preconceptional maternal body mass index (BMI), maternal country of birth, collective daycare attendance at one year of age, and number of siblings in household at one year of age. For the present study, the “human milk consumption” and “duration of human milk consumption” variables referred to any human milk intake whatever the source (own mother or donor), the mode (bottle or breast), and exclusively or not. Moreover, cohort specific factors were investigated: the antibiotherapy during labor and during the last hospitalization leading to delivery (referred to as antibiotherapy during pregnancy) were included for the EPIPAGE 2 cohort; the age at complementary feeding (first introduction of food other than milk or water), the age at first introduction of different food (vegetables, fruits, meat and fish), antibiotics administration during the first year of life, and rural or urban environment of parental dwelling at 2 months were included for the ELFE cohort. Additionally, we included information regarding maternal diet in the last 3 months of pregnancy though adherence to the French dietary intake guidelines (Programme National Nutrition Santé (PNNS)) [14]. We used a score of adherence to general guidelines for adults (PNNS score) and a score of adherence to guidelines specific to pregnant women (pregnancy score) [15].

For each factor, we designed a multivariate model to adjust for a set of parsimonious potential confounding factors identified with a Directed Acyclic Graph (DAG). We adopted a thorough adjustment scheme considering several confounding factors while being careful not to adjust for potential mediating factors.

Minimal adjustment sets identified for each factor were as follows:-**gestational age**: maternal age, maternal level of education, household income and, preconceptional maternal BMI.-**delivery mode**: gestational age, maternal age, maternal level of education, household income, country of birth of the mother and preconceptional maternal BMI.-**early neonatal antibiotics**: delivery mode, gestational age, maternal level of education and skin-to-skin contact.-**human milk consumption**: maternal age, maternal level of education, household income, country of birth of the mother, gestational age, delivery mode and household siblings at 1 year.-**skin-to-skin contact**: Country of birth of the mother, gestational age and human milk consumption.-**preconceptional maternal BMI**: country of birth of the mother.-**daycare exposure at 1 year**: maternal level of education, household income, country of birth of the mother and household sibling at 1 year.-**household sibling at 1 year and environment (rural/urban)**: maternal age, maternal level of education and household income.-**country of birth of the mother**: none.-**antibiotherapy during pregnancy**: none-**antibiotherapy during delivery**: antibiotherapy during pregnancy.-**age at complementary feeding + age at first introduction** of vegetables, fruits, meat, and fish: maternal age, maternal level of education, household income, country of birth of the mother, preconceptional maternal BMI, human milk consumption, daycare exposure at 1 year and household siblings at 1 year.-**PNNS and pregnancy scores**: maternal age, maternal level of education, country of birth of the mother, household income, preconceptional maternal BMI, environment (rural/urban), and household siblings at 1 year.-**antibiotherapy during the first year of life**: maternal age, maternal level of education, daycare exposure at 1 year and household siblings at 1 year.

### 2.6. Samples Collection, DNA Extraction, Sequencing and Data Processing

Stool samples were collected at home circa 3.5 years of age, according to the operating procedure of the International Human Microbiome Standards (IHMS) [16]. Briefly, stools were collected in a stabilizing solution (RNA later ^®^) and sent to the study biobank. Within 3 days, samples were homogenized, aliquoted, and stored at −80 °C until analysis. Total fecal DNA was extracted according to the IHMS operating procedure [16]. Negative (water) and positive (microbial mock communities) controls were included and went through the same process as samples from extraction to sequencing. High throughput sequencing of the V3-V4 regions of the 16s ribosomal RNA gene was performed on an Illumina MiSeq platform as previously described [17]. Raw sequences were analyzed using the pipeline “Find Rapidly OTU with Galaxy Solution” (FROGS, v 3.2) from the Galaxy software framework [18]. Sequences were checked for quality using FastQC (threshold 20–38). After trimming barcodes, the sequences were filtered for length and clustered into operational taxonomic units (OTUs) using the swarm clustering method implemented in the FROGS version 3.2 pipeline. OTUs representing less than 0.005% of all the sequences, the majority corresponding to singleton OTUs, were discarded [19]. Chimera were detected using vsearch (v 2.17) implemented in FROGs pipeline and then removed. Taxonomic affiliation of OTUs was assigned using Silva 138.1 pintail100 database. A total of 24,698,345 reads (median 29,240 reads per sample) were obtained from the 16S rRNA gene sequencing. To adjust for the influence of uneven sampling depth, each sample was rarefied to the minimum sampling depth of the dataset (9258 reads) and rarefied data were used for all downstream analyses unless stated otherwise.

### 2.7. Microbiota Diversity and Composition

Metrics used to describe the gut microbiota included alpha diversity, beta diversity and enterotypes. Chao1 estimate (richness), and Shannon index (evenness) were calculated to investigate patterns of microbial community alpha diversity in the gut microbiota. Beta diversity was assessed by computing dissimilarity matrices using Bray–Curtis and Unweighted Unifrac (UniFrac) distances. The Bray–Curtis dissimilarity distance reflects community composition considering the abundance of taxa, while the UniFrac distance considers the phylogenetic relationships among members of the bacterial communities. Stratification of the cohort based on gut microbial composition was performed according to previously reported enterotyping guidelines [20]. Briefly, the relative abundances of the classified genera were used to produce a Jensen–Shannon divergence (JSD) distance matrix between samples. Based on the obtained distance matrix, samples were clustered using the Partitioning Around Medoids (PAM) algorithm. The optimal number of clusters was chosen by maximizing the Calinski–Harabasz index and was cross-validated with the silhouette index and prediction strength. The result of PAM clustering was visualized on PCoA biplot, and driver genera vectors were assessed using the “envfit” function. The genus with the highest relative abundance in each group was considered the main contributor of each enterotype

### 2.8. Statistical Analysis

Associations between the different factors and alpha diversity metrics and enterotypes were tested by multivariable linear or logistic regression, respectively, adjusted for different sets of confounding factors presented in section “Exposures of Interest” according to the factor tested. For beta diversity indices, hypothesis testing was carried out using permutational multivariate analysis of variance [21] (PERMANOVA) adjusted for appropriate confounding factors and based on 999 permutations to analyze the influence of the factors on the gut microbiota according to the beta diversity metrics. For all PERMANOVA analyses, Benjamini–Hochberg’s FDR (False Discovery Rate) correction was used to adjust for multiple testing.

To identify single taxonomic units associated with specific factors using differential abundance testing methods, we used a triangulation of three different methods: Linear Decomposition Models (LDM) [22]. Analysis of Compositions of Microbiomes with Bias Correction (ANCOM-BC) [23], and ANOVA-Like Differential Expression tool for compositional data (ALDEx2) [24]. For each factor, the model was controlled for its own set of confounding factors and was corrected using the Benjamini–Hochberg method. Differences in abundance were based on non-rarefied counts for both ANCOM-BC and ALDex2 as both methods used their own normalization methods.

Sensitivity analyses were conducted by excluding all children from the EPIPAGE 2 cohort (*n* = 172) in order to assess the influence of very preterm status on the associations. We also performed a comparative analysis to assess the impact of missing data. Missing values in the exposures were imputed using a random forest-based algorithm for missing data imputation called missForest [25]. Results of these sensitivity analyses are reported as Appendix A.

All analyses were performed using the R software version 4.1.2 (R Foundation) and the following R packages: phyloseq (v1.34.0), vegan (v2.6-0), ggplot2 (v3.3.5), lme4 (v1.1-27.1), cluster (v2.1.2), clusterSim (v0.49-2), fpc (v2.2-9), ade4 (v1.7-16), missForest (v1.5), LMD (v4.0), ANCOM-BC (v2.1.0), and ALDEx2 (v1.28.1).

## 3. Results

Stool samples of children included in the present study were collected at 3.5 years of age corresponding to a median age of 42.3 months (interquartile range [IQR], 41.4–44.0 months) and 42.6 months (IQR, 42.2–43.3 months) for ELFE and EPIPAGE 2 children, respectively.

Characteristics of ELFE, EPIPAGE 2, and pooled study population are provided in Appendix A. The gestational age (GA) distribution of the pooled study population is as follows: 4.3% (*n* = 34) children within the 24–26 GA range; 17.3% (*n* = 138) within the 27–32 GA range; 2.8% (*n* = 22) within the 33–37 GA range; 74.1% (*n* = 591) ≥ 37 GA.

### 3.1. Gut Microbiota Description of Children at 3.5 Years of Age

At 3.5 years of age, the gut microbiota of children from the EPIPAGE 2 cohort harbored lower alpha diversity compared to the ELFE children (Chao1 estimate, *p* = 0.004; Shannon index, *p* ≤ 0.001) (Figure 1A). The gut microbiota of the 798 children optimally stratified into two enterotypes equally distributed between ELFE and EPIPAGE 2 children (Figure 1B,C). Each enterotype was characterized by the dominance of either *Bacteroides* (B_type, *n* = 649) or *Prevotella* (P_type, *n* = 149) (Figure 1D) with P_type enterotype harboring higher richness (*p* ≤ 0.001, Appendix A).

### 3.2. Enterotypes Association with Early Life Factors

In adjusted analyses (Table 1), having ever received human milk, attendance of a childcare center at one year, and having at least one sibling during the first year of life were associated with higher probabilities of belonging to the P_type enterotype. Late introduction of fish (>6 months) was associated with a lower probability of belonging to the P_type enterotype. The same tendency was observed for late introduction of meat (>6 months). A trend was also observed for a negative association between administration of antibiotics to mothers during delivery and P_type enterotype.

### 3.3. Early Life Factors Influencing Gut Microbiota Diversity at 3.5 Years of Age

#### 3.3.1. Alpha Diversity

Perinatal factors such as gestational age, mode of delivery, and having a sibling at home during the first year of life were the most significantly associated factors with the gut microbiota alpha diversity (Table 2). Children born with lower gestational age showed lower richness without difference in term of diversity. Children born by caesarean section showed less richness and diversity. Children who had at least one sibling were characterized with increased richness and diversity. Human milk consumption was not associated with the gut microbiota alpha diversity but when considering this factor as a continuous variable, a tendency toward a higher diversity for longer duration of human milk consumption was observed; however, the effect size was small, and the association failed to reach significance (Table 2). Of note, the median duration of human milk consumption was 2.46 months (IQR, 0.23–5.98 months). Having a sibling at home during the first year of life was the factor most significantly associated with the gut microbiota as children who had at least one sibling were characterized with increased richness and diversity compared to single children. Regarding maternal factors, higher richness and diversity were observed in children gut microbiota born to overweight and obese mothers respectively. Additionally, children whose mothers were born outside of France were characterized by lower diversity without difference in richness (Table 2).

#### 3.3.2. Beta Diversity

Regarding the different factors associated with the gut microbiota beta diversity, according to both Bray–Curtis and UniFrac dissimilarity distances, having a sibling at home during the first year of life explained the greatest amount of variance between subjects (Figure 2A,B). Gestational age was the second factor most significantly associated with the beta diversity followed by the mode of delivery according to both distances. Daycare exposure during the first year of life was only associated with the bacterial composition according to the Bray–Curtis distance. Maternal factors including maternal preconceptional BMI and the country of birth were associated with the composition of the gut microbiota at 3.5 years of age according to the UniFrac distance.

### 3.4. Differential Abundance Testing

Because identifying taxonomic units associated with specific factors using differential abundance testing methods can produce heterogeneous results [26], we implemented a triangulation of three different methods to assess differential abundance of taxa at both family and genus levels. Among the factors tested, the gestational age, household siblings, and the country of birth of the mother were associated with differential abundance of taxa according to the three methods. Compared to full-term children, the gut microbiota of preterm children was characterized with a lower abundance of six genera, and a higher abundance of one unclassified genus belonging to the *Ruminococcaceae* family (Figure 3A). Having at least one sibling during the first years of life was associated with lower abundance of 10 genera and higher abundance of 8 genera (Figure 3B). The country of birth of the mothers was also associated with differential abundance: children from mothers born in France showed lower abundance of an unclassified genus from the *Oscillospiraceae* family compared to children from mother born outside of France. At the family level, being born preterm was associated with lower abundance of *Butyricicoccaceae, Erysipelatoclostridiaceae* and *Bifidobacteriaceae* families and increase abundance of the *Oscillospiraceae* family. Having at least one sibling was associated with a decreased abundance of *Bacteroidaceae, Veillonellaceae, Eggerthellaceae* and *Bifidobacteriaceae* families and an increased abundance of *Marinifilaceae, Christensenellaceae* and *Barnesiellaceae* families (Appendix A).

### 3.5. Sensitivity Analysis

We showed that gestational age still imprinted the gut microbiota at 3.5 years of age. To check the sensitivity of our analysis, we re-runed it on a dataset containing only ELFE children (*n* = 626) to verify if the associations described above were also observed in moderately preterm and full-term infants or driven by the very and extremely preterm subgroups. Overall, we found consistent associations when we only considered the ELFE children (Appendix A). However, gestational age was not associated with the gut microbiota anymore, as expected by the lesser gestational age variability in the ELFE sample. Moreover, the delivery mode was not associated with the beta diversity, but the associations still remained regarding the alpha diversity. The same genera were found to be differentially abundant in ELFE children gut microbiota with at least one sibling compared to only children for the exception of *Dorea*, *Barnesiella* and *Tyzzerella.* At the family level, the gut microbiota of ELFE children with at least one sibling was differentially less abundant in *Enterobacteriaceae* and the differences in the *Barnesiellaceae* and *Veillonellaceae* families were no longer existent (Appendix A).

Missing data did not impact the overall associations found in this study as consistent direction of associations were observed with imputed data (Appendix A). Nonetheless, the significant association found between the P_type enterotype and the late introduction to fish was no longer significant (Appendix A).

## 4. Discussion

Considering different metrics to describe the gut microbiota of children enrolled in two well described French national birth cohorts, we showed that several early life factors including gestational age, delivery mode, human milk consumption, maternal overweight, and household composition are associated with the gut microbiota of children at 3.5 years of age.

Among the different perinatal factors investigated, gestational age was one of the most significantly associated with the gut microbiota composition. Gestational age had a significant effect on the gut microbiota beta diversity and children born with higher gestational age showed a higher alpha diversity. Our results based on 798 children support those of Fouhy et al. [27] reporting that gestational age imprints the gut microbiota up to 4 years of age (*n* = 70 participants). Additionally, we found that the gut microbiota of preterm children was characterized with lower levels of multiple genera among which the *Bifidobacterium* genus. Recently, it was suggested that a *Firmicutes*-dominant stage precedes the *Bifidobacterium*-dominant stage during early life enterotype transitions, particularly in preterm infants, that is strongly associated with gut microbiota immaturity at 1 year of age [28]. Moreover, the lower abundance of *Bifidobacterium* has been observed in preterm children at 2 years of age [29]. These data, consistent with our findings, suggest that the initial delay of colonization of *Bifidobacterium* found in preterm neonates [30] may represents a long-lasting effect of prematurity. Members of the *Bifidobacterium* genus have beneficial effects on the host, [31] and reduce levels of bifidobacteria have been proposed to be associated with potential diseases in childhood [32]. The lower levels of *Bifidobacterium* observed in preterm children may be one contributing mechanism involved in the higher risk of obesity [33] or neurodevelopmental outcomes [34] during childhood observed in this particular population.

In the present study, the delivery mode was associated with the gut microbiota overall composition assessed by the beta diversity. However, this association was no longer existent when excluding preterm children from the EPIPAGE 2 cohort. This may be explained by the higher number of cesarean sections among preterm children (Appendix A). Nonetheless, children born by cesarean section showed a decreased alpha diversity, independently of preterm status. This is consistent with previous results [35] indicating that the effect of delivery mode is still noticeable during early childhood. Cesarean delivery is reported to increase the risk of later asthma and overweight/obesity in childhood that can be related to this long-term gut microbiota dysbiosis [36]. In recent years, one recurring pattern associated to host health status is a greater richness and diversity at the structural and functional levels [37]. The loss of diversity being described as one type of dysbiosis, [38] it led to propose the gut microbiota as a potential target to prevent future pediatric detrimental outcomes.

Breastmilk contains beneficial factors including immunoglobulin A, lactoferrin, defensins [6], and oligosaccharides which modulate innate immunity, intestinal cell responses, and the gut microbiota [39]. Introduction to solid food is followed by a decrease in breastmilk intake and the change in diet to solid food results in major shifts in microbial community composition [7] unless breastfeeding is continued [8,40]. In the present study, we observed an absence of effects of human breastmilk intake regarding alpha and beta diversities. However, the children gut microbiota was characterized by two discrete enterotypes dominated by either *Bacteroides* or *Prevotella* in line with our previous findings on a smaller sample size [17] and other studies in children [28,41,42,43]. We showed that children who had ever received human milk were associated with the P_type enterotype independently of the intake duration. Despite the widespread accepted belief that breastfeeding has a prominent impact on the gut microbiota during early infancy (first year of life) [8,11], our data suggest an extended influence of human milk consumption on shaping microbial composition. These data support the positive association between breastfeeding and *Prevotella* abundance in older children population (6–10 years) [44]. During maturation of the gut microbiota associated with cessation of breastmilk and introduction to solid food, *Bifidobacterium* abundance tends to decrease, allowing new microbial community to settle in order to degrade new substrates introduced by solid food consumption [8]. *Prevotella*-driven enterotype is associated with rich-fiber diet and intake of carbohydrates and simple sugars [45]. However, if the higher prevalence of P_type among children who have ever received human milk is caused by the human milk consumption itself or other dietary factors is unclear. *Prevotella*-driven enterotype in adolescents [46] or elevated *Prevotella* abundance in adults [47] have been previously associated with the Mediterranean diet, which is considered as one of the healthiest dietary patterns. Previous findings showed that longer exclusive breastfeeding duration was associated with a healthier child diet at 3 years of age [48] and that children aged 2–8 years who had been breastfed in infancy were associated with a healthy dietary pattern in childhood compared to never breastfed children [49]. Therefore, the association between the human milk consumption and the P_type enterotype may be explained by a higher adherence to a healthy dietary pattern in children who had ever received human milk. However, in our study, maternal diet through adherence to French dietary intake guidelines potentially modeling children dietary pattern was not associated with the P_type enterotype. Additionally, we found that early introduction to fish (≤6 months) was associated with the P_type enterotype. This association completely disappeared in sensitivity analyses with imputed data, making this association to interpret with caution.

Environmental factors describing the milieu in which children grew during their first year showed effects on the child gut microbiota at 3.5 years. Children with at least one sibling present in the household were characterized by a higher alpha diversity and an overall differential composition assessed by the beta diversity compared to their single-child counterparts. Previous studies showed that living with a sibling was associated with the gut microbiota during childhood [8,50]. In line with our results, Laursen et al. [50] reported that children living with siblings were characterized with a higher alpha diversity. In our study, the presence of a sibling was associated with differential abundance of 18 genera, including a positive association with strict anaerobes within the *Bacteroidetes* phylum (*Odoribacter* and *Barnesiella*) previously associated with having older siblings at 18 months [50]. The ratio of strict to facultative anaerobes 12 months after birth was shown to be lower in single children than in children with older siblings, suggesting the theory of strict anaerobic bacteria acquisition from older siblings. An association between the presence of siblings and *Assacharobacter* and *Gordonibacter* genera has also been proposed [50]. Another study found a positive correlation between *Bifidobacterium* and sibship size at 1 year of age [51]. In the present study, the association with having a sibling at one year may also be partially mediated by the number of siblings at 3.5 years as this number is likely higher in those that had already at least one sibling at 1 year. Interestingly, the P_type enterotype was also associated with children having siblings and daycare center attendance. In concordance with our findings, previous studies showed elevated abundance of *Prevotella* genus in children attending daycare at 1 year of age [52] or in children with older siblings with the effect persisting up to 6 years of age [53]. It is likely that the presence of siblings in the household or attending daycare centers are more likely to affect the number of different bacteria to which an infant is exposed, affecting richness and diversity even though, no difference of alpha diversity was observed for daycare attendance. The *Prevotella*-driven enterotype has been previously linked to lower risk of inflammatory bowel disease in adult populations, [54,55] and type 1 diabetes in children, [56] and *P. copri*, a major *Prevotella* species of the P_type enterotype has been recently associated with a general healthy state [55]. Altogether, these data support the hypothesis that the *Prevotella*-driven enterotype may be a potential biomarker of human health related state [55].

Geographical location can have an impact on the gut microbiota as previous research reported differences in gut microbiota development in children up to 3 years of age [8,28]. In our study, we showed that maternal country of birth was associated with the beta diversity and that children born to mothers born outside of France harbored lower alpha diversity. While we did not assess directly the geographical location of the child, maternal origin of birth could be related to specific dietary habits or cultural factors that can shape the children gut microbiota. Nonetheless, the majority of our population of mothers were born in France (91.5%). The remaining 8.5% of mothers born outside of France were from European union and northern Africa countries. Given the lack of subjects within each remaining categories, differences between mothers born outside of France could therefore not be assessed.

Vertical transmission of maternal bacteria is the primary source of gut colonization in neonates [6] and obesogenic transfers of microbiota from mother to child may be a mechanism to explain the transgenerational transmission of obesity risk [57]. In the present study, maternal preconceptional BMI was associated with the children gut microbiota beta diversity. Additionally, children born to overweight/obese mothers showed an increased gut microbiota alpha diversity. These results are consistent with some but not all [58] previous findings showing such associations in a 2-year-old children population, although, the effect was only seen in the higher socioeconomic status group [59]. The study of Tun et al. [60] showed same patterns with higher alpha diversity observed among infants aged 3–4 months born to obese mothers independently of the delivery mode. Contrary to our study, these previous studies also demonstrated differential abundance of specific genus associated with maternal prepregnancy obesity. We further sought to test the hypothesis of obesogenic transfers of microbiota from mother to child by testing the association between maternal preconceptional BMI and the gut microbiota diversity stratified by delivery mode. We found an absence of significant interaction between maternal preconceptional BMI and delivery mode in our population (p_interaction:Chao1_ = 0.92 and p_interaction:Shannon_ = 0.80). Nonetheless, the higher alpha diversity observed in children born to overweight/obese mothers is somewhat at odds with the hypothesis of diversity as a marker of healthy status as these children are more likely to become overweight themselves later in childhood [60]. Although, maternal prepregnancy BMI can particularly affect child early life gut microbiota, other related lifestyle factors acting later in life such as diet and physical activity may mediate the observed associations.

### Study Limitation and Strengths

Working on a large population, we chose to use the DNA extraction protocol validated by the IHMS consortium for its high reproducibility; however, it tended to underestimate the proportion of Gram-positive bacteria and potentially the diversity of the gut microbiota composition [16]. Nonetheless, despite this limitation, other studies found similar results as our study as discussed throughout the manuscript while using different DNA extraction protocols.

The use of the 16S rRNA gene sequencing approach limits this study to the description of bacterial composition. Analysis using shotgun metagenomics would allow greater sequencing depth, thus providing additional information regarding microbiome functional profiles and meaning at the species level of the microbiome. Because gut microbiota analysis was only performed at 3.5 years of age, we could not assess the longitudinal effect of the early life factors on the development of the gut microbiota. Further longitudinal cohort studies are needed to validate our findings and assess differences in the longitudinal development of the gut microbiota according to specific factors such as gestational age for example (term vs. preterm infants).

Major strengths of this study include the recruitment of a large population of French children from two well described birth cohorts providing an accurate description of the early life factors. Particularly, inclusion of a large sample of preterm children rarely studied at this age allow more variation and helped to decipher the lack of data regarding this population. Most of the studies in the literature examining the effects of early life factors on the gut microbiota usually focus on a limited number of factors. Another strength of this study is the analysis of several early life factors all at once in a single study which allows comparing their effects without the distortion of different methods and study power. We carefully adjusted multivariable models taking into account potential confounding factors. We were careful not to adjust for potential mediators. This allows a correct estimation of the total effect of the considered early life factors on gut microbiota characteristics. We do not disregard the fact that the observed effects can be in part mediated by other factors operating after the first year of life in a causal pathway but whatever the underlying mechanisms, modulation early in life of the studied factors has the potential to influence the child microbiota at 3.5 years of age.

## 5. Conclusions

Based on full-term- and preterm children’s national cohorts, our findings support the evidence of the influence of early life factors on the gut microbiota in preschool children at 3.5 years of age independently of preterm status, even though, effect of prematurity on the gut microbiota was still noticeable. From perinatal factors to the environmental living conditions through diet and maternal health, we provide solid arguments in favor of early life exposures as important determinants in the shaping of the future adult-like gut microbiota at a pivotal time of microbial dynamics. Our data support the importance of the first three years of life as a critical window for the development of the gut microbiota and future child health. The prevalence of factors able to negatively influence the gut microbiota composition has increased and may have contributed to the rise of non-communicable diseases. Our findings merit further studies to investigate the influence of microbiota-targeted health-promoting strategies early in life.

## Figures and Tables

**Figure 1 microorganisms-11-01390-f001:**
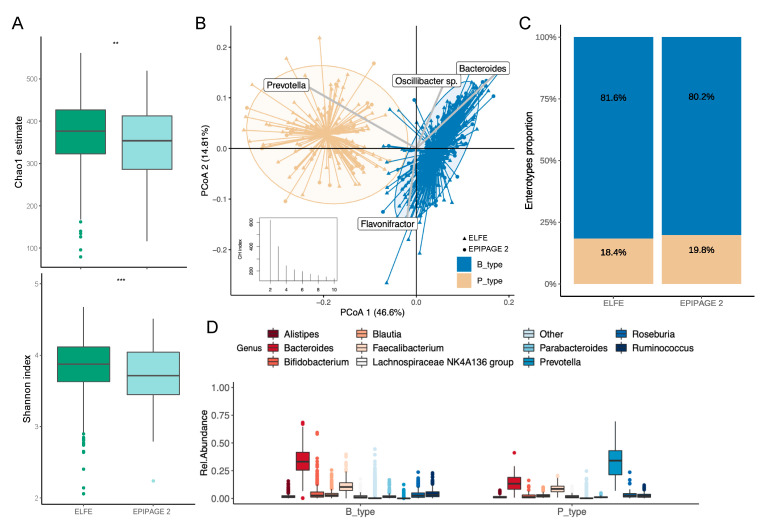
Gut microbiota composition and enterotypes of the children at 3.5 years of age. Groups are colored according to their enterotypes. (**A**) Boxplot of alpha diversity assessed by Chao1 estimate (top) and Shannon index (bottom) based on the OTU taxonomic profile between ELFE and EPIPAGE 2 children (** *p*  ≤ 0.01; *** *p* ≤ 0.001). The boxplots show the smallest and largest values, 25% and 75% quartiles, the median, and outliers. (**B**) Clustering based on the genus taxonomic profile. Biplot arrows indicate the top four genera that drive samples to different locations on the plot. Calinski–Harabasz index showing the optimal number of clusters is plotted. (**C**) Distribution of enterotypes among ELFE and EPIPAGE 2 children. The distribution is expressed as proportion of each enterotype among each group of children. (**D**) The boxplots represent the relative abundance of the top 10 genera distributed among the P_ and B_type enterotypes. The boxplots show the smallest and largest values, 25% and 75% quartiles, the median, and outliers.

**Figure 2 microorganisms-11-01390-f002:**
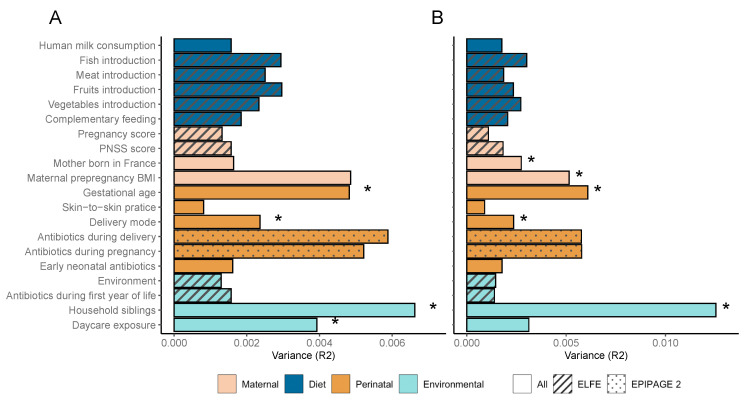
Association between early life factors and gut microbiota beta diversity. Horizontal bars show the amount of variance (R2) explained by each factor in its own model analyzed using PERMANOVA. The groups within each factor are further detailed in (Appendix A). Factors are colored based on overall metadata group and patterned regarding the availability of the covariate among the two cohorts. Significance of factors (FDR *p* adjusted ≤ 0.05) are denoted with an asterisk. (**A**) Associations using Bray–Curtis distance. (**B**) Associations using UniFrac distance.

**Figure 3 microorganisms-11-01390-f003:**
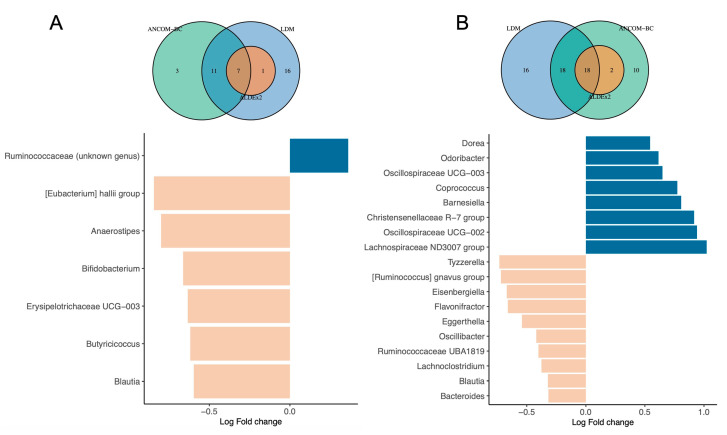
Differential abundance genus associated with gestational age and household siblings. Ven diagrams at the top of each barplot illustrate the number of differentially abundant genera identified by each method as well as overlaps of the three different methods. Barplots show the ANCOM—BC log-transformed in abundance results at the genus level. Barplots are colored based on positive or negative Log Fold change in abundance. Results from the other methods at both genus and family levels are presented as Appendix A. (**A**) Differential abundance testing for gestational age (<37 GW vs. ≥37 GW). (**B**) Differential abundance testing for household siblings (At least one vs. none).

**Table 1 microorganisms-11-01390-t001:** Multivariate association between children enterotypes and early life factors.

		Enterotype P_type vs. B_type		
	N	OR ^a^	95% CI	*p*-Value
**Pooled sample**				
**Gestational age (weeks)**	708	1.02	0.97, 1.08	0.39
**Delivery mode**	697			0.61
Vaginal		—	—	
Cesarean		0.88	0.53, 1.43	
**Early neonatal antibiotics**	704			0.75
No		—	—	
Yes		0.87	0.38, 1.97	
**Human milk consumption**	632			**0.015**
No		—	—	
Yes		1.99	1.14, 3.68	
**Duration of human milk consumption (months)**	632	1.01	0.97, 1.05	0.60
**Skin-to-skin practice**	695			0.51
Yes		—	—	
No		0.81	0.41, 1.52	
**Maternal prepregnancy BMI**	777			0.34
Underweight		0.90	0.40, 1.82	
Normal		—	—	
Overweight		1.17	0.71, 1.87	
Obese		1.75	0.92, 3.18	
**Daycare exposure at 1 year**	722			**<0.001**
Family		—	—	
Childcare assistant/Paid home help		0.69	0.43, 1.10	
Daycare centre		2.05	1.15, 3.65	
**Household siblings at 1 year**	728			**0.045**
No		—	—	
At least 1		1.51	1.01, 2.28	
**Mother born in France**	798			0.46
Yes		—	—	
No		1.26	0.67, 2.25	
**ELFE**				
**Age at complementary feeding**	493			0.83
≤6 months		—	—	
>6 months		1.08	0.52, 2.14	
**Age at first vegetables introduction**	492			0.45
≤6 months		—	—	
>6 months		0.79	0.42, 1.43	
**Age at first fruits introduction**	493			0.24
≤6 months		—	—	
>6 months		0.72	0.40, 1.24	
**Age at first meat introduction**	483			0.092
≤6 months		—	—	
>6 months		0.60	0.33, 1.09	
**Age at first fish introduction**	477			**0.019**
≤6 months		—	—	
>6 months		0.45	0.24, 0.88	
**Maternal pregnancy diet (PNSS score) ***	560			0.17
≤7.80		—	—	
>7.80		1.37	0.87, 2.17	
**Maternal pregnancy diet (Pregnancy score) ***	560			0.43
≤7.75		—	—	
>7.75		1.21	0.76, 1.93	
**Antibiotics in the first year**	609			0.17
Yes		—	—	
No		0.74	0.47, 1.14	
**Environment**	607			0.37
Rural		—	—	
Urban		1.24	0.78, 2.03	
**EPIPAGE 2**				
**Antibiotics during delivery**	149			0.072
No		—	—	
Yes		0.41	0.14, 1.08	
**Antibiotics during pregnancy**	149			0.79
No		—	—	
Yes		1.13	0.47, 2.70	

Logistic regression models. Abbreviations: N = number of children included in each model; P_type, enterotype enriched in *Prevotella*; B_type, enterotype enriched in *Bacteroides*; CI, Confidence Interval; BMI, Body Mass Index; PNSS, Programme National Nutrition Santé. ^a^ Each covariate is adjusted for its own set of parsimonious confounding factors presented in the “Exposures of interest” section. * Chosen cut points are median scores within the population. *p*-values marked with bold indicate statistically significant *p*-values (*p* ≤ 0.05).

**Table 2 microorganisms-11-01390-t002:** Multivariate association between gut microbiota alpha diversity and early life factors.

		Chao1 Estimate			Shannon Index		
Pooled Sample	N	Beta ^a^	95% CI	*p*-Value	Beta ^a^	95% CI	*p*-Value
**Gestational age (weeks)**	708	1.8	0.27, 3.3	**0.021**	0.00	0.00, 0.01	0.31
**Delivery mode**	697			**0.004**			**<0.001**
Vaginal		—	—		—	—	
Cesarean		−22	−37, −7.1		−0.12	−0.20, −0.05	
**Early neonatal antibiotics**	704			0.28			0.11
No		—	—		—	—	
Yes		14	−11, 40		0.10	−0.02, 0.23	
**Human milk consumption**	632			0.41			0.79
No		—	—		—	—	
Yes		6.4	−8.9, 22		0.01	−0.08, 0.06	
**Duration of human milk consumption (months)**	632	0.52	−0.69, 1.7	0.40	0.01	0.00, 0.01	0.076
**Skin-to-skin practice**	695			0.89			0.56
Yes		—	—		—	—	
No		−1.4	−21, 19		0.03	−0.07, 0.13	
**Maternal prepregnancy BMI**	777			**0.007**			**0.039**
Underweight		9.7	−12, 32		0.05	−0.06, 0.16	
Normal		—	—		—	—	
Overweight		22	6.6, 37		0.10	0.02, 0.17	
Obese		26	4.7, 48		0.08	−0.02, 0.19	
**Daycare exposure at 1 year**	722			>0.99			0.92
Family		—	—		—	—	
Childcare assistant/Paid home help		0.30	−14, 14		0.00	−0.07, 0.07	
Daycare centre		−0.66	−20, 19		0.02	−0.11, 0.08	
**Household siblings at 1 year**	728			**<0.001**			**<0.001**
No		—	—		—	—	
At least 1		37	25, 49		0.16	0.10, 0.22	
**Mother born in France**	798			0.24			**0.047**
Yes		—	—		—	—	
No		−12	−32, 8.2		−0.10	−0.20, 0.00	
**ELFE**							
**Age at complementary feeding**	493			0.95			0.59
≤6 months		—	—		—	—	
>6 months		−0.63	−22, 21		0.03	−0.08, 0.14	
**Age at first vegetables introduction**	492			0.92			0.46
≤6 months		—	—		—	—	
>6 months		0.92	−17, 19		0.03	−0.06, 0.13	
**Age at first fruits introduction**	493			0.92			0.83
≤6 months		—	—		—	—	
>6 months		0.80	−15, 17		0.01	−0.07, 0.09	
**Age at first meat introduction**	483			0.43			0.74
≤6 months		—	—		—	—	
>6 months		−7.1	−25, 11		0.02	−0.07, 0.11	
**Age at first fish introduction**	477			0.52			0.91
≤6 months		—	—		—	—	
>6 months		−6.7	−27, 13		0.01	−0.11, 0.10	
**Maternal pregnancy diet (PNNS score) ***	560			0.74			0.54
≤7.80		—	—		—	—	
>7.80		2.2	−11, 15		0.02	−0.04, 0.08	
**Maternal pregnancy diet (Pregnancy score) ***	560			0.78			0.69
≤7.75		—	—		—	—	
>7.75		1.9	−11, 15		0.01	−0.05, 0.08	
**Antibiotics in the first year**	609			0.33			0.11
Yes		—	—		—	—	
No		6.5	−6.4, 19		0.05	−0.01, 0.12	
**Environment**	607			0.75			0.72
Rural		—	—		—	—	
Urban		−2.3	−16, 12		0.01	−0.08, 0.06	
**EPIPAGE 2**							
**Antibiotics during delivery**	149			0.50			0.72
No		—	—		—	—	
Yes		12	−23, 46		0.03	−0.13, 0.19	
**Antibiotics during pregnancy**	149			0.99			0.33
No		—	—		—	—	
Yes		−0.23	−33, 33		0.08	−0.08, 0.23	

Linear regression models. Abbreviations: N = number of children included in each model; CI, Confidence Interval; BMI, Body Mass Index; PNSS, Programme National Nutrition Santé. ^a^ Each covariate is adjusted for its own set of parsimonious confounding factors presented in the “Exposures of interest” section. * Chosen cut points are median scores within the population. *p*-values marked with bold indicate statistically significant *p*-values (*p* ≤ 0.05).

## Data Availability

Personal data of children from the ELFE and EPIPAGE2 cohorts cannot be made publicly available for ethical reasons. They are available upon reasonable request from the authors under data-security conditions. The 16S rRNA gene reads are publicly available from the National Center for Biotechnology Information (nih.gov) Sequence Read Archive (SRA) under the Bioproject accession number PRJNA907285.

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
