# Peer review of "Early Life Factors Influencing Children Gut Microbiota at 3.5 Years from Two French Birth Cohorts"

_microorganisms, 2023, doi:10.3390/microorganisms11061390_

Round 1

Reviewer 1 Report

The authors analyzed the association between 20 early life factors and their effect on gut microbiota in children at the age of 3.5 years. The study is well conducted with a good sample size and included several early life factors in a single study. The results support that early life factors influence gut microbiota independently of preterm status. Microbial richness and diversity are critical for the child's health and the outcomes of this study can help in designing strategies to promote healthy microbiota in children early on.

There are a few questions related to this study that can influence its impact on a larger population.

1.     The current study is focused on French birth cohorts and found that gestational age was one of the most associated factors. What could be the relation between a particular ethnicity in relation to microbial richness and diversity?

2.     The association between these factors and gut microbiota could be different based on the geographical location of the mother or/and children. Did the authors find or studied anything along these lines?

Author Response

The authors analyzed the association between 20 early life factors and their effect on gut microbiota in children at the age of 3.5 years. The study is well conducted with a good sample size and included several early life factors in a single study. The results support that early life factors influence gut microbiota independently of preterm status. Microbial richness and diversity are critical for the child's health and the outcomes of this study can help in designing strategies to promote healthy microbiota in children early on.

We are thankful for the reviewer comments.

There are a few questions related to this study that can influence its impact on a larger population.

  1. The current study is focused on French birth cohorts and found that gestational age was one of the most associated factors. What could be the relation between a particular ethnicity in relation to microbial richness and diversity?

Different geographical locations are related to specific dietary habits, cultural, climatic and economic factors and can indeed, shape the gut microbiota development during the early stages of life. For example, the large TEDDY cohort followed 900 children from birth up to 3 years and showed some taxa prevalence differences in the gut microbiota development between 3 and 22 months of life while no differences in term of diversity have been found up to 3 years of age. Of note this study included children from westernized and developed countries including the USA, Finland, Sweden and Germany (Stewart et al., 2018). Another study including 1950 children from 17 country covering 6 continents from birth up to 3 years of age showed that among developing countries such as India, Bangladesh, South Africa, Peru, and Brazil, children were characterized with a more immature gut microbiota throughout the first 3 years of life compared to developed countries in Northern Europe, such as Finland, Norway, and Estonia (Xiao et al., 2021). Although this study did not specifically investigate microbial richness and diversity, it suggests that gut microbiota development can be shaped by geographic location and all environmental factors that are associated with it such as cultural difference and dietary habits. Particularly, it is known that breastfeeding and its duration are important for the gut microbiota development and children breastfed tend to have a lower gut microbiota diversity. Breastfeeding can depend on geographic and cultural differences as for example, in France, breastfeeding duration is short.

  1. The association between these factors and gut microbiota could be different based on the geographical location of the mother or/and children. Did the authors find or studied anything along these lines?

In the present study, we found that maternal country of birth (France vs other) has an impact on the children overall gut microbiota composition (beta diversity) and that children born to mothers born outside of France were characterized with lower alpha diversity (Shannon index). As stated above, we agree that the effects of some factors on the children gut microbiota may be associated with geographical location and more precisely ethnicities such as age as duration of breast milk intake or age at complementary feeding. In our two French birth cohorts, ELFE and EPIPAGE 2, all children were born in France and majority of our population of mothers were born in of France (91.5%). The remaining 8.5% of mothers born outside of France were from European union and northern Africa countries. Given the lack of subject within each remaining categories, differences between mothers born outside of France could therefore not be assessed.

However, through a large cohort of infants, our study brings solid arguments that early life exposures still impact the gut microbiota at 3.5 years when microbiota has acquired many characteristics of the adult microbiota. Such impact of early exposures is certainly similar regardless of the geographical location, although the exposures involved may differ according to environment and lifestyle.    

Taking into accounts the reviewer comments, we added in the manuscript a small discussion paragraph about the maternal origin of birth lines 473-483 : “Geographical location can have an impact on the gut microbiota as previous research reported differences in gut microbiota development in children up to 3 years of age [8], [28]. In our study, we showed that maternal country of birth was associated with the beta diversity and that children born to mothers born outside of France harbored lower alpha diversity. While we did not assess directly the geographical location of the child, maternal origin of birth could be related to specific dietary habits or cultural factors that can shape the children gut microbiota. Nonetheless, the majority of our population of mothers were born in France (91.5%). The remaining 8.5% of mothers born outside of France were from European union and northern Africa countries. Given the lack of subjects within each remaining categories, differences between mothers born outside of France could therefore not be assessed.

Reviewer 2 Report

In overall, the manuscript is well written, except that 'breast milk' is two words rather than one. I'd encourage the authors to use 'human milk' rather than 'breast milk'.

To my understanding, the study investigated factors that influence the gut microbiota of preterm and term delivery children at the age of 3.5. However, there is a huge difference in sample size between the EPIPAGE (preterms) and ELFE (terms) cohorts. I'm afraid that the statistical power might not be strong enough for the comparison between the two groups. For instance, EPIPAGE cohort harboured lower alpha diversity compared to the ELFE children, and could this be due to the 4 times smaller sample size of EPIPAGE? Gestation age may be significantly influences the gut microbiome of newborns, but I doubt that this is the case for children. The proper way to examine if the gestation age influence the gut microbiome of children is to perform a longitudinal study of preterm and term babies up to the age of 3.5. I'd advise the authors to carefully interpreting the results of gestation age (Line 375-378), and focusing on the investigation of other factors in EPIPAGE and in ELFE (not based on pooled EPIPAGE and ELFE samples). 

It'd be great to change Table 1 to charts, in order to have a clear visualisation how the address factors influence the proportion of enterotype P-type and B-type.

Please check if the Bioproject accession number is correct because it is not found. 

Author Response

In overall, the manuscript is well written, except that 'breast milk' is two words rather than one. I'd encourage the authors to use 'human milk' rather than 'breast milk'.

We are thankful for the reviewer comments. According to the comment, we replaced the term “breastmilk” by “human milk” throughout all the manuscript.

To my understanding, the study investigated factors that influence the gut microbiota of preterm and term delivery children at the age of 3.5. However, there is a huge difference in sample size between the EPIPAGE (preterms) and ELFE (terms) cohorts. I'm afraid that the statistical power might not be strong enough for the comparison between the two groups. For instance, EPIPAGE cohort harboured lower alpha diversity compared to the ELFE children, and could this be due to the 4 times smaller sample size of EPIPAGE? Gestation age may be significantly influences the gut microbiome of newborns, but I doubt that this is the case for children. The proper way to examine if the gestation age influence the gut microbiome of children is to perform a longitudinal study of preterm and term babies up to the age of 3.5. I'd advise the authors to carefully interpreting the results of gestation age (Line 375-378), and focusing on the investigation of other factors in EPIPAGE and in ELFE (not based on pooled EPIPAGE and ELFE samples). 

We demonstrated a significant effect on the gestational age (as a continuous variable and not in groups) on the gut microbiota alpha diversity at 3.5 years, therefore our analysis does not lack statistical power. Additionally, when we re-runned the analysis excluding the EPIPAGE2 children (preterm children n = 172) we showed no significant effect of gestational age anymore (supplementary data, table S4) suggesting that there is a real effect of this smaller group on the gut microbiota diversity at 3.5 years. Furthermore, we previously found a difference in alpha diversity of the gut microbiota at 3.5 years of age between EPIPAGE 2 and ELFE children in a more balanced design (ELFE = 200 and EPIPAGE 2 = 159) (doi : 10.3389/fmicb.2022.919317). While this previous finding did not take into account confounding factors, in the present study, we took into account important confounder, therefore strengthening our previous findings. Besides, our data are in agreement with the results of Fouhy et al. (doi.org/10.1038/s41467-019-09252-4), indicating that gestational age still imprint the gut microbiota up to 4 years of age (Fouhy et al., 2019).

We used the pooled samples as our primary analysis because we wanted to assess as much variation from the larger sample size. We performed the analysis in ELFE children only (supplementary data, tables S3-S6) and found similar results as in our primary pooled analyses (except for gestational age).

It'd be great to change Table 1 to charts, in order to have a clear visualisation how the address factors influence the proportion of enterotype P-type and B-type.

We agree that a chart would have been useful to have a clear visualization of the prevalence of each enterotype according to the factors. However, we investigated some factors as continuous variables such as gestational age (in weeks) or breast milk duration (in months). Therefore, plots visualization cannot be made due to the heterogeneity of the variables format (both categorical and continuous) and it is more appropriate to present our Table 1 as the results of our regressions models.

Please check if the Bioproject accession number is correct because it is not found.

The Bioproject accession number is correct but cannot be found yet. The public release of the data will be made upon publication of the manuscript. Here is the reviewer link:

https://dataview.ncbi.nlm.nih.gov/object/PRJNA907285?reviewer=drri16rf9oc8bfbpk2eiugejrf